# Cutaneous α-Synuclein and Age Spots in Neurodegeneration: A Systematic Review and Testable Hypothesis

**DOI:** 10.3390/ijms27010096

**Published:** 2025-12-22

**Authors:** Arianna Di Stadio, Pietro De Luca, Beatrice Francavilla, Massimo Ralli, Stefano Di Girolamo, Iole Indovina, Giulia Ciccarese, Laura Dipietro

**Affiliations:** 1Life and Health Science Department, Link University, 00165 Rome, Italy; 2IRCCS Santa Lucia, Laboratory of Neuromotor Physiology, IRCCS Santa Lucia Foundation, 00179 Rome, Italy; 3Otolaryngology Department Santa Rosa Hospital, 01100 Viterbo, Italy; 4Otolaryngology Department, University of Rome Tor Vergata, 00133 Rome, Italy; 5UNICAMILLUS, International Medical University, 00131 Rome, Italy; 6Department of Systems Medicine and Centre for Space BioMedicine, University of Rome Tor Vergata, 00133 Rome, Italy; 7Dermatology Department, University of Foggia, 71122 Foggia, Italy; 8Highland Instruments, Inc., Cambridge, MA 02238, USA; lauradp@highlandinstruments.com

**Keywords:** alpha-synuclein, phosphorylated alpha synuclein, skin, Parkinson’s Disease, neurodegeneration, neuroinflammation, age spots

## Abstract

Recent studies have identified phosphorylated α-synuclein in the skin of individuals with neurodegenerative disease. The levels of this protein in the skin have been correlated with disease severity. This protein has been specifically studied in alpha-synucleinopathies such as Parkinsons’ Disease (PD) and Multiple System Atrophy (MSA). Given that skin biopsy studies have often shown high levels of phosphorylated α-synuclein in the neck area, and that other clinical studies have described easily identifiable changes in facial skin features, this systematic review explores whether age spots, which are very common in sun-exposed areas (hands and face), could serve as early indicators of neurodegenerative disease. We performed a systematic review of the literature in three databases (Google, Scopus and Pubmed) following PRISMA guidelines. We used the following keywords: “alpha-synuclein and skin”, “alpha-synuclein and skin spots”, “alpha-synuclein and solar lentigo”, “alpha-synuclein and age spots”, “alpha-synuclein and melanocytes”, “skin biopsy and synucleinopathies”, “skin biopsy and neurodegenerative disease”, and “Parkinson’s Disease. Eleven studies were identified and included. Based on the study results and preliminary evidence in the literature evaluating the effect of α-synuclein on keratinocytes and melanocytes, we propose that the accumulation of this protein within the skin may produce visible alterations that can be quantified to enable early, noninvasive detection of neurodegenerative disease.

## 1. Introduction

The skin and brain share a common embryological origin from the ectodermal germ layer [1]. This ancestral relationship may underlie several processes, including the effects of aging on both organs, manifesting as cognitive decline in the brain and structural changes in the skin, such as wrinkles, laxity, and age spots [2]. Age spots, also called solar lentigines or lentigo senilis, are light-brown to black lesions of various sizes that frequently develop in chronically ultraviolet (UV)-exposed skin (face, dorsum of the hands, upper trunk, and bald scalp) [2,3]. Because of these shared developmental pathways, the skin represents an accessible peripheral tissue for exploring mechanisms of brain aging and neurodegeneration [4].

α-Synuclein is predominantly localized at presynaptic terminals, where it associates with synaptic vesicle membranes and regulates neurotransmitter release. Its precise physiological role remains debated, as both inhibitory and facilitatory effects on synaptic transmission have been reported. In Parkinson’s Disease (PD) and related synucleinopathies, misfolded α-synuclein aggregates recruit monomeric α-synuclein and propagate between neurons, leading to progressive dysfunction at synaptic, cellular, and systems levels [5]. Beyond the central nervous system, α-synuclein is ubiquitously expressed and occurs at higher levels in melanoma. This overlap suggests that factors related to melanin and α-synuclein, such as oxidative stress and inflammation, autophagic dysfunction, and protein aggregation, may play a role in both conditions [6,7].

A recent multicenter study demonstrated that phosphorylated α-synuclein can be detected in skin punch biopsies from normal-appearing sites, including the posterior cervical (neck) site, of patients with synucleinopathies, with high positivity rates (93–100% across PD, Dementia with Lewy Bodies (DLB), Multiple System Atrophy (MSA), and Pure Autonomic Failure (PAF)) and rare positives in healthy controls (~3%) [8]. Interestingly, even in non-synucleinopathies such as Alzheimer’s disease (AD), cutaneous inflammatory and microvascular abnormalities [2] have been reported, along with altered dermal pH [4]. However, because up to 50% of AD patients can present alpha-synuclein pathology, and AD is primarily considered in relation to tauophathy and amyloidopathy, we excluded this condition from our review. The findings that have been observed in synucleinopathies suggest that peripheral skin changes may occur in association with neurodegenerative processes of diverse etiologies.

Earlier work [9] detected increased epidermal α-synuclein immunoreactivity in the skin of patients with PD and melanoma, while skin tags showed no increase [9]. However, that study was limited by a small sample size (<100), heterogeneous conditions (healthy subjects, PD, melanoma, nevi, and skin tags), and the use of non-phospho-specific (total) α-synuclein immunohistochemistry [9]. Notably, the authors did not examine or classify age-related pigmented lesions, which is a significant limitation, given their biological variability [9].

To date, no study has examined the potential influence of α-synuclein on age spots, a typical cutaneous sign of aging or differentiated between typical aging-associated spots and those potentially linked to neurodegenerative pathology. This systematic review, therefore, aims to explore whether specific characteristics of these spots reflect underlying α-synuclein-related processes. Our working hypothesis is that α-synuclein interferes with the normal skin aging process [10], not by increasing melanin production, but by altering melanocyte function and pigment distribution, and by amplifying the normal inflammatory process of aging [10,11,12]. Such mechanisms, known to occur in skin aging and pigmentary disorders [10,11], could modify the persistence, pigment distribution, and morphology of age spots, yielding lesions that are more persistent and irregularly pigmented than typical age spots, thereby providing a potential noninvasive window into neurodegenerative disease. In line with this possibility, Jucevičiūtė et al. [12] hypothesized, and in a small pilot study found, suggestive associations that certain skin and hair features may emerge as preclinical signs of PD, supporting the broader view that cutaneous phenotypes can mirror neurodegenerative processes.

## 2. Results

Figure 1 shows the literature search process. Table 1 summarizes the studies identified through the search strategy and included based on the predefined inclusion and exclusion criteria. No studies on α-synuclein and skin spots/age spots/solar lentigo were identified.

A total of 11 studies met the inclusion criteria after applying the predefined search and screening process. 6 studies were performed in the US, 1 included multiple countries, and the remaining were from the UK (n = 1), Italy (n = 1), Lithuania (n = 1), and Mexico (n = 1). Most studies were cross-sectional diagnostic investigations, with a few retrospective chart reviews and one exploratory case–control study. The combined sample comprised 976 participants, of whom 136 were healthy controls (Figure 2A). The total number of patients was 840, of whom 7 had melanoma. The remaining 833 had synucleinopathies or related conditions, including 337 (40.4%) with PD, 149 (17.9%) with unspecified α-synucleinopathy, 126 (15.1%) with MSA, 76 (9.1%) with DLB, 36 (4.3%) with type 1 Gaucher Disease, 33 (4.0%) with PAF, 28 (3.4%) with rapid eye movement sleep behavior disorder (RBD), 18 (2.2%) with Progressive Supranuclear Palsy (PSP), 12 (1.4%) with non α-synuclein-related autonomic failure, 10 (1.2%) with atypical Parkinsonism, and 8 (1.0%) with Corticobasal Syndrome (CBS) (Figure 2B).

## 3. Discussion

The data extracted from the literature (Table 1) show that α-synuclein is detectable in the skin of individuals with synucleinopathies and helps distinguish affected patients from healthy controls. In several cohorts, it also differentiates among synucleinopathy subtypes (e.g., PD vs. MSA vs. PAF). As the review progressed, the search strategy was refined to also encompass terms related to skin pigmentation and aging (e.g., “skin spots”, “age spots”, “solar lentigo”) to explore potential visible cutaneous manifestations of α-synuclein accumulation. However, no studies directly addressing α-synuclein in age spots were identified. The collected data identified the presence of α-synuclein in patients with neurological diseases, but without any information about the presence of clinically visible skin spots in the individuals included in the study. For this reason, we can only speculate about a possible relationship between skin spots and α-synuclein. We therefore considered related evidence linking α-synuclein to skin physiology, melanocyte biology, and pigmentation to inform a hypothesis about visible age-related skin changes.

By contrast, a recent dermatofluoroscopic study found no difference in melanin spectral profiles (shape or intensity) between patients with PD and healthy controls when examining normo-pigmented skin [21]. Another study reported increased epidermal α-synuclein immunopositivity in PD skin versus controls, with even higher levels in nevi and the highest in melanoma; notably, the nevi and melanoma samples were from non-PD cohorts [9]. The contrasting findings between normo-pigmented (normal skin) and pigmented lesions (melanoma) [6,9,21] highlight the need for further research to determine whether visible cutaneous features could serve as biomarkers of α-synucleinopathies. To date, the literature has not established whether α-synuclein accumulation produces any specific or detectable skin changes.

Identifying a “visual parameter” for early detection of neurodegenerative conditions is a clinically important goal. Here, “early” refers to the stage around or shortly after clinical diagnosis, when cutaneous α-synuclein pathology is already detectable, but prior to advanced neurodegenerative progression. This definition does not imply a prodromal or pre-symptomatic phase, but rather an early stage within clinically manifest disease. In this context, skin spots, more precisely age spots, may represent a promising opportunity, as their development involves melanocyte and keratinocyte activity [3,22,23,24,25], and these cell types seem to be extremely sensitive to the effects of α-synuclein [26] (Figure 3).

The heterogeneous findings from studies comparing normal skin and melanoma in patients with α-synucleinopathies and healthy individuals drew our attention to age spots.

Age spots are characterized by a reduced renewal of suprabasal keratinocytes and the persistence of pigment [3,22,23,24] within melanocytes, with their onset and persistence associated with a micro-inflammatory environment [25]. We therefore hypothesize that α-synuclein may modulate visible pigmentary features, particularly age spots.

Multiple lines of evidence support the biological plausibility of our hypothesis. Researchers have reported elevated α-synuclein in melanoma tissue and higher epidermal α-synuclein in the skin of patients with PD [9]. In 2012, Pan et al. demonstrated that ultraviolet B (UVB) exposure induced melanin synthesis in A375 and SK-MEL-28 melanoma cells, while α-synuclein expression attenuated this UVB-induced increase in melanin synthesis, suggesting that α-synuclein may exert inhibitory effects on melanin production [27]. Rachinger et al. [26] showed that α-synuclein alters melanocyte morphology and is required for efficient melanosome release and transfer to keratinocytes, without affecting melanin synthesis. Oliveira et al. showed that oligomeric α-synuclein reduces keratinocyte proliferation by triggering NF-κB–mediated inflammation, characterized by increased IL-1β and TNF-α expression. The antiproliferative effect was partially reversed by an IL-1β inhibitor or dexamethasone, whereas TNF-α inhibition had no effect [28].

The microphthalmia-associated transcription factor (MITF), a master regulator of melanocyte differentiation, positively regulates α-synuclein expression [26]. α-Synuclein localizes to melanosomes and facilitates their transport and release without altering melanin synthesis, implying that α-synuclein dysregulation could change pigment distribution independent of pigment production [26].

Collectively, the evidence from melanoma studies, together with experimental findings in melanocytes and keratinocytes exposed to α-synuclein, supports the notion that α-synuclein–related alterations in skin cells may underlie observable pigmentary changes. Further research is warranted to determine whether such alterations could be leveraged for noninvasive screening of synucleinopathies.

### Proposed Mechanistic Hypothesis: α-Synuclein and Pigmentary Changes in Aging Skin

Based on the evidence reviewed, we propose that the biological effects of α-synuclein on epidermal cells could manifest as visible changes in common age-related skin spots. In other words, cellular alterations induced by α-synuclein might lead to distinct “α-synuclein-associated” skin spots.

α-Synuclein is a pro-inflammatory protein in keratinocytes: exogenous oligomeric α-synuclein has been shown to trigger NF-κB–mediated inflammation with increased IL-1β and TNF-α, leading to reduced cell proliferation and epidermal thinning [28].

In melanocytes, α-synuclein localizes to melanosomes, where it contributes to normal organelle dynamics and pigment transfer; silencing α-synuclein alters melanocyte morphology and reduces melanosome release and transfer to keratinocytes, while total melanin content remains unchanged [26]. Under UVB, α-synuclein also attenuates tyrosinase activity stimulation in melanoma cells, thereby reducing melanin synthesis [27]. Together, these data suggest that α-synuclein can modulate both melanosome trafficking [26] and, under UVB, enzymatic regulation of melanogenesis via tyrosinase attenuation [27]. Biochemical work further shows that α-synuclein interacts directly with tyrosinase, supporting a possible α-syn–tyrosinase cross-talk [29]. Moreover, a recent review highlights dysregulated human tyrosinase as a putative link between pigmentation pathways and neurodegeneration [30], further supporting the biological plausibility of such crosstalk. Importantly, while these studies support the mechanistic plausibility of α-synuclein–tyrosinase interactions, they do not yet establish any direct effects on age-related skin spots.

Building on these observations, we hypothesize that α-synuclein, by regulating melanosome trafficking and positioning without increasing total melanin levels [26], could alter pigment packaging and spatial distribution, leading to lighter or irregularly toned cutaneous areas. In individuals with α-synucleinopathy, such effects might manifest as age spots that tend to appear paler overall, with uneven, clumped pigment that increases local contrast, i.e., paler and more mottled than common age spots (Figure 4).

Although α-synuclein may alter melanosome formation and distribution [26], the fundamental process of melanin transfer from melanocytes to keratinocytes likely remains operative, even if the specific routes are heterogeneous and context-dependent [31].

Finally, considering the effects of α-synuclein on both keratinocytes and melanocytes [26,28], we hypothesize that the combined impact (melanosome accumulation within a thinner epidermis due to reduced keratinocyte turnover and potentially altered degradative pathways such as autophagy) could further modify skin pigmentation. α-Synuclein-associated spots might therefore appear lighter in color (primarily due to altered melanosome distribution and epidermal thinning, while, under UVB exposure, α-synuclein may also attenuate melanin synthesis [27]) and less uniform (due to altered melanosome distribution) compared to common age spots; moreover, α-synuclein-associated spots may be less prone to regression than melasma [31,32]. Under dermoscopy, they might show irregular pigment clumping, with sharper borders but not darker in tone. Moreover, because of chronic IL-1-mediated inflammation [28], α-synuclein-related spots might present an erythematous halo or irregular margins compared with regular age spots.

To reiterate, this work proposes a theoretical framework that integrates available evidence on age spots with studies describing the effect of α-synuclein at the cellular level on the keratinocytes and melanocytes. To date, no studies have compared age spot characteristics between elderly individuals without α-synucleinopathies and those affected by such conditions; therefore, this mechanism remains hypothetical.

Nonetheless, the hypothesis opens a new and testable research avenue. Comparative histological studies could assess differences in skin spots between healthy individuals and patients with PD, DLB, or MSA. Advanced imaging approaches that visualize melanin and melanosome distribution (e.g., the M-INK–based confocal technique [33]) could help compare typical age-related spots with putative α-synuclein–associated spots. Furthermore, image-based analyses supported by Artificial Intelligence could potentially detect differences in spot colors due to the accumulation of hypopigmented melanosomes under thinner skin in people with α-synucleinopathies. The combination of high melanosome density and thin epidermis might give the spots a different appearance from common age spots; the expectation is that these spots are paler than typical aging spots and have more irregular borders.

## 4. Materials and Methods

This study was conducted in accordance with the Preferred Reporting Items for Systematic Reviews and Meta-Analyses (PRISMA) guidelines (Figure 1). Ethical approval from the Institutional Review Board was not required for this type of review.

### 4.1. Search Strategy

Two researchers (PDL and MR), under the supervision of ADS, performed a comprehensive literature search on PubMed, Scopus, and Google Scholar for papers in the English language published between 2015 and 2025. The search strategy combined MeSH terms and free-text keywords, including “alpha-synuclein and skin”, “alpha-synuclein and skin spots”, “alpha-synuclein and solar lentigo”, “alpha-synuclein and age spots”, “alpha-synuclein and melanocytes”, “skin biopsy and synucleinopathies”, “skin biopsy and neurodegenerative disease”, and “Parkinson’s Disease”. Boolean operators (AND, OR) were applied strategically to refine the searches, ensuring sensitivity and specificity.

Abstracts of all retrieved articles were independently screened, and duplicate entries were identified and systematically removed using reference management software. The remaining articles were cross-checked for consistency, and any disagreement regarding study inclusion or exclusion was resolved through structured discussions and iterative consensus meetings involving all three researchers (PDL, MR, and ADS). To ensure methodological transparency and reproducibility, the systematic review adhered strictly to the PRISMA guidelines, with a PRISMA flow diagram constructed to track the selection process. Additionally, the reference lists of all included studies were manually reviewed to identify further relevant articles that may have been overlooked in the database searches.

### 4.2. Study Selection Criteria

The inclusion and exclusion criteria were pre-specified to ensure a transparent and rigorous selection process.


*Inclusion Criteria:*
Studies providing quantitative data on characteristics of skin in neurodegenerative disease in humans.Studies published in peer-reviewed journals.Studies involving adult patients with α-synucleinopathy.Articles published in English



*Exclusion Criteria:*
Studies involving children or adolescents under 18 years old.Studies conducted on animals.Systematic reviews, case reports, and commentaries.Non-peer-reviewed articles, conference abstracts, or opinion pieces.


### 4.3. Data Extraction

Data extraction was performed independently by two researchers (ADS and PDL) to ensure accuracy and minimize bias. A structured data extraction form was developed and piloted before use to standardize the process. For each study, the following information was systematically recorded: author(s), year of publication, study design, country of origin, sample size, participant demographics, and results of the study. To maintain data integrity, all extracted information was cross-verified by the supervising investigator (ADS) during biweekly review meetings. Any discrepancies identified during data extraction were resolved through discussion or, when necessary, by consulting an external expert in systematic review methodology.

The extracted data were organized in a comprehensive spreadsheet for quantitative synthesis and qualitative analysis. This approach facilitated the identification of emerging patterns, research gaps, and potential sources of bias across the included studies.

### 4.4. Risk of Bias Assessment

The National Institutes of Health (NIH) Quality Assessment Tools for Case–Control Studies were used to assess the risk of bias, as these checklists are designed for various study designs (National Heart, Lung, and Blood Institute. Study Quality Assessment Tools. (2014), available at: https://www.nhlbi.nih.gov/health-topics/study-quality-assessment-tools; accessed on 19 December 2025). Each study was categorized as poor, fair, or good quality. Incomplete data were considered a source of bias, and studies were rated fair even if they were unbiased and fully described. Two authors independently scored each article, and disagreements were resolved by consensus.

## 5. Conclusions

The studies currently available in the literature have identified the presence of α-synuclein in the skin of individuals affected by α-synucleinopathies. However, we did not find any studies specifically examining skin spots in patients with α-synucleinopathies or other neurodegenerative diseases. Currently, skin biopsy remains the only reliable method for detecting the presence and accumulation of this protein in the skin, even though a dermatofluoroscopic study did not find any difference in the cutaneous melanin spectral profile (shape or intensity) between patients with PD and healthy controls when examining normo-pigmented skin [21].

However, skin biopsy is an invasive procedure and considering that the highest levels of phosphorylated α-synuclein have often been observed in the skin of the neck [8], its use as a large-scale screening tool for detection at early stages of α-synucleinopathies may have limited acceptance.

Although our literature review did not identify studies addressing skin spots, the proposed mechanistic hypothesis of the role that α-synuclein might have in skin cells provides a rationale for conducting targeted investigations on skin spots. Herein, we propose that the evaluation of paler, more irregularly pigmented spots could represent a potential noninvasive peripheral marker of neuroinflammatory neurodegeneration. After controlling for potential confounders, such as hormonal influences [34] or medications [35], these spots could provide insights into underlying neurodegenerative processes. This hypothesis integrates dermatological evidence on age spots with cellular studies describing the effects of α-synuclein on keratinocytes and melanocytes.

We believe that studies bridging dermatology and neurology should be encouraged, especially given the strong embryological relationship between the skin and central nervous system and emerging evidence linking cutaneous alterations to neurodegenerative disorders.

## Figures and Tables

**Figure 1 ijms-27-00096-f001:**
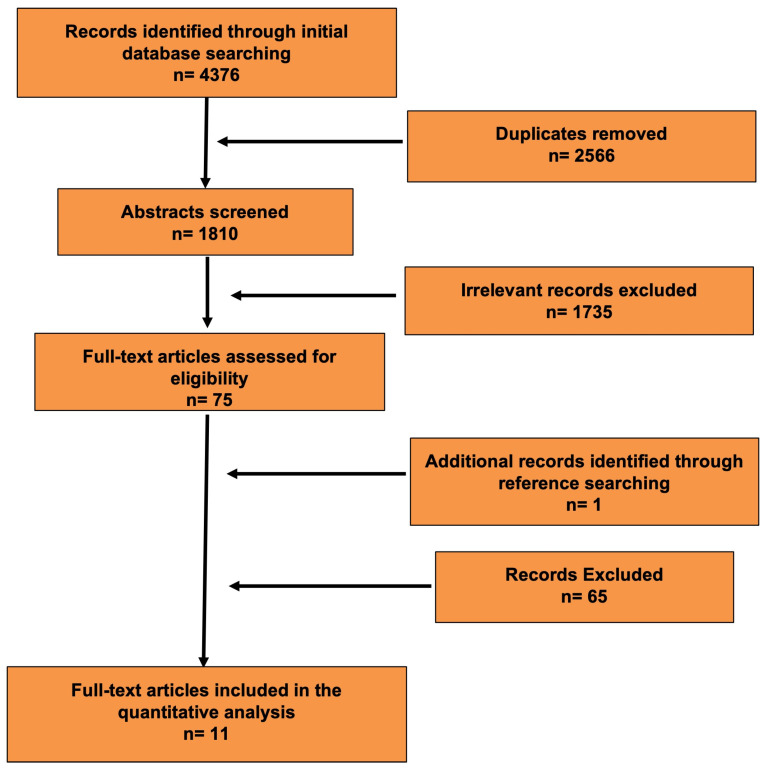
Literature search process (Preferred Reporting Items for Systematic Reviews and Meta-Analyses flow diagram).

**Figure 2 ijms-27-00096-f002:**
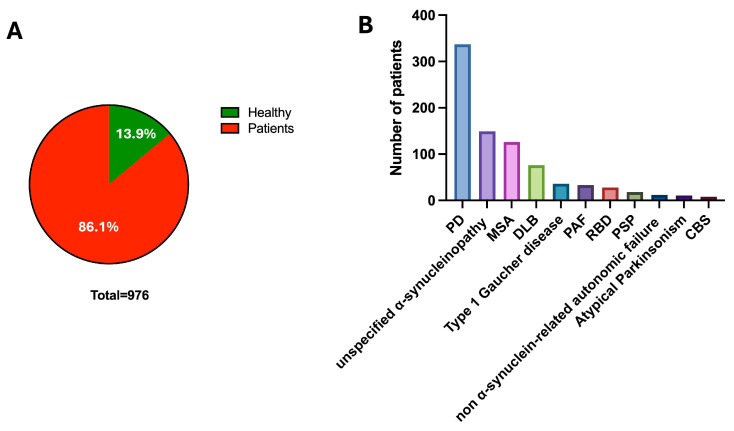
(**A**) The pie chart shows the number of controls and patients included in the study. (**B**) The graph shows the distribution of patients in the sample analyzed, excluding those with melanoma (7), for a total of 833 individuals with neurological diseases.

**Figure 3 ijms-27-00096-f003:**
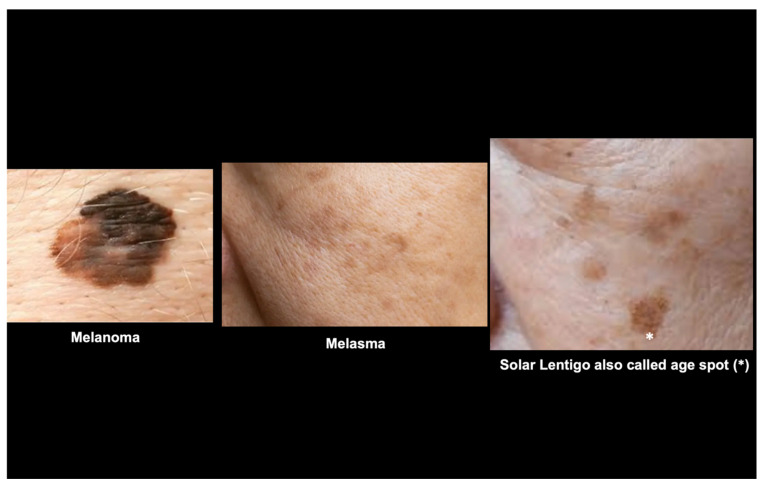
The image illustrates the difference among pigmented lesions that can be observed in melanoma, melasma, and age spots/solar lentigo (*).

**Figure 4 ijms-27-00096-f004:**
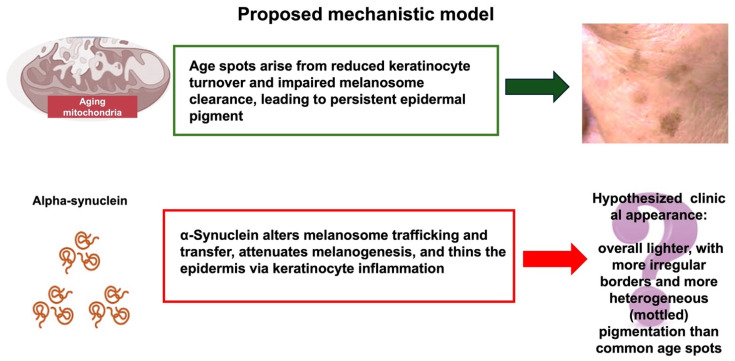
The image summarizes a mechanistic hypothesis that might explain differences between aging skin spots and α-synuclein-associated skin spots.

**Table 1 ijms-27-00096-t001:** Summary of studies examining pathologic α-synuclein in the skin of individuals with neurodegenerative diseases.

Author, Year, Country	Type of Study	Sample (n)	Age *	Disease	Findings	Quality Rating
Rodriguez-Leyva et al., Mexico, 2017 [9]	Cross-sectional	48	49.9	PD (n = 8) melanoma (n = 7)	Patients with PD and melanoma have increased staining for α-synuclein in their skin compared to controls.	Fair
Jucevičiūtė et al., Lithuania, 2019 [12]	Exploratory cross-sectional case–control pilot	32	35	PD (n = 32)	PD is associated with earlier age at onset of hair greying, greater tendency to sunburn, difficulty tanning and non-normal skin type.	Good
Wang et al., United States, 2020 [13]	Diagnostic accuracy study with retrospective (post-mortem) and prospective (ante-mortem) components	160	68	PD (n = 47) MSA (n = 3) DLB (n = 7)	Skin αSynP aggregation seeding activity may serve as a novel biomarker for ante-mortem diagnosis of PD and other synucleinopathies.	Good
Giannoccaro et al., Italy, 2022 [14]	Cross-sectional diagnostic case–control	52	71	PSP (n = 18) CBS (n = 8) PD (n = 26)	All PD patients, and only 2/26 with a clinical diagnosis of PSP/CBS, had skin P-SYN deposits. Furthermore, these two patients had clinical features suggesting an atypical synucleinopathy presentation or a mixed pathology.	Good
Al-Qassabi et al., Multicenter, 2021 [15]	Cross-sectional diagnostic case–control; autopsy assay-development phase	58	67.3	PD (n = 20) RBD (n = 28) atypical Parkinsonism (n = 10)	Even with a single 3 mm punch biopsy, there is considerable promise for using P-SYN deposition in skin to diagnose both clinical and prodromal PD.	Fair
Gibbons et al., United States, 2023 [16]	Prospective cross-sectional	85	62	PD (n = 54) MSA (n = 31)	All patients with MSA and 51/54 with PD had evidence of P-SYN in at least one skin biopsy. No P-SYN was detected in controls. Patients with MSA had greater P-SYN deposition (*p* < 0.0001) and more widespread peripheral distribution (*p* < 0.0001) than patients with PD. These combined measures achieved ~97% sensitivity and 98% specificity in distinguishing between the 2 disorders.	Good
Koay et al., United Kingdom, 2025 [17]	Prospective cross-sectional (diagnostic)	36	61	PAF (n = 11) MSA (n = 13) Non-synucleinopathy-related autonomic failure (n = 12)	Cutaneous P-SYN was abundant in PAF, a predominantly peripheral α-synucleinopathy. It is a promising biomarker to help distinguish PAF from MSA and non-synucleinopathy-related autonomic failure, aiding early diagnosis and recruitment to future clinical trials.	Fair
Isaacson et al., United States, 2024 [18]	Retrospective chart review	97	71	PD (n = 54) MSA (n = 24) DLB (n = 19)	In patients with suspected synucleinopathies, skin biopsy detection of P-SYN demonstrated high clinical utility, leading to changes in clinical diagnosis and treatment.	Fair
LoPiccolo et al., United States, 2024 [19]	Cross-sectional	36	59.3	GD1 *	10 participants (27.8%) were αSyn SAA positive, 7 (19.4%) were intermediate, and 19 (52.8%) were negative. αSyn SAA positivity was associated with older age (*p* = 0.043), although αSyn SAA positivity was more prevalent in patients with GD1 than historic controls.	Fair
Gibbons et al., United States, 2024 [8]	Blinded cross-sectional (diagnostic)	223	69.5	PD (n = 96) MSA (n = 55) DLB (n = 50) PAF (n = 22)	A high proportion of individuals meeting clinical consensus criteria for PD, DLB, MSA, and PAF had P-SYN detected by skin biopsy.	Good
Gummerson et al. United States, 2025 [20]	Retrospective chart review	149	72.7	α-synuclein-related disease (n = 149)	Out of 149 biopsies, 105 (70%) had ≥1 positive skin sample. No pre-biopsy symptom was significantly associated with a positive biopsy.	Fair

PD, Parkinson’s Disease; MSA, Multiple System Atrophy; DLB, Dementia with Lewy Bodies; αSynP, pathological α-synuclein; PSP, Progressive Supranuclear Palsy; CBS, Corticobasal Syndrome; P-SYN, phosphorylated α-synuclein; RBD, Rapid Eye Movement Sleep Behavior Disorder; PAF, Pure Autonomic Failure; GD1, Type 1 Gaucher Disease; * included because it confers an increased risk of developing PD; αSyn SAA, α-synuclein seeding amplification assay. “*” Mean age in years, including both patients and controls.

## Data Availability

Original data come from literature and are fully available for all readers.

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
