# Peer review of "Cutaneous α-Synuclein and Age Spots in Neurodegeneration: A Systematic Review and Testable Hypothesis"

_ijms, 2025, doi:10.3390/ijms27010096_

Round 1
Reviewer 1 Report
Comments and Suggestions for Authors
This is an interesting study that points to future research directions, but it is incomplete. My reservations concern the number of nine authors in such a short study. I don't understand why Alzheimer's disease was omitted from the search. The exclusion of animal studies in the exclusion criteria is also incomprehensible. I also noted several editorial errors. In Figure 1, as many as 2,566 duplicates were removed – this is incomprehensible. Table 1 should be reworded and some of the data from the description above should be placed below the table. Column names should be made simpler and more compact, not descriptive. The introduction should mention neurodegeneration following cerebral ischemia and experimental changes in the synuclein gene in the interesting, recently published paper Folia Neuropathol. 2025;63(2):120-126
Comments on the Quality of English LanguageFor publication after suggested changes will be done.
Author Response
We thank the reviewers for the careful reading of our manuscript and for the valuable comments, which helped us clarify several aspects that were missed or insufficiently explained in our original submission.
We have carefully considered all comments and revised the main text accordingly, using track changes to highlight the modifications.
In the document below, our point-to-point responses to the reviewers’s comments are in bold, and the passages shown in bold Italicscorrespond to text that has been directly incorporated into the revised manuscript.
We hope that the revisions and clarifications provided have satisfactorily addressed all concerns and have strenghtened the manuscript.
Reviewer 1
This is an interesting study that points to future research directions, but it is incomplete.
My reservations concern the number of nine authors in such a short study.
One author had been incorrectly included and has been removed. The remaining 8 authors are members of the investigative team that conducted our systematic review.
I don't understand why Alzheimer's disease was omitted from the search.
Alzheimer's disease (AD) was not included because we limited our review to primary α-synucleinopathies. Although up to 50% of AD cases can present alpha-synuclein pathology, AD is mostly considered a tauophaty and amyloidopathy, characterized by the accumulation of amyloid-beta (Aβ) plaques and neurofibrillary tangles composed of hyperphosphorylated tau protein. We have clarified this point in the Introduction.
The exclusion of animal studies in the exclusion criteria is also incomprehensible.
We focused our attention on human studies because our goal was to evaluate whether cutaneous α-synuclein could serve as a biomarker in humans. As such, animal studies were not relevant to our research purpose and were therefore excluded.
I also noted several editorial errors. In Figure 1, as many as 2,566 duplicates were removed – this is incomprehensible.
We used three databases (Google, PubMed, and Scopus) as detailed in the Materials and Methods section (pages 5-6). For this reason, the number of duplicates was high.
Table 1 should be reworded and some of the data from the description above should be placed below the table.
We have moved some of the information previously contained in the column titles to the bottom, as you suggested.
Column names should be made simpler and more compact, not descriptive.
We have simplified the column titles while maintaning their meaning.
The introduction should mention neurodegeneration following cerebral ischemia and experimental changes in the synuclein gene in the interesting, recently published paper Folia Neuropathol. 2025;63(2):120-126
We reviewed this interesting article; however, we do not think its content aligns with the aim of our manuscript. The referenced study focuses on gene changes and alpha-synuclein accumulation in the brain, whereas our work examines the effects of this protein in the skin. Given that the biological actions of alpha-synuclein differ across tissues (PMCID: PMC3446957 DOI: 10.1371/journal.pone.0045183) we believe that this article, although valuable, should not be included in our review.
Reviewer 2 Report
Comments and Suggestions for Authors
The topic is interesting and of differential diagnostic relevance in early α-synucleinopathies.
Abstract: Alzheimer is not topic of the manuscript and should therefore not be highlighted in the abstract. Skin changes in sun-exposed areas of the skin are addressed in the abstract, but findings in relation to PD and other α-synucleinopathies found in the literature search are not described but hypothesized. The abstract does not sufficiently reflect and summarize the contents of the manuscript.
Manuscript: The methods of the literature research one (Table 1) are not described (pubmed? Scopus? Google?...Key words?). Not described are the “predefined criteria” of inclusion and exclusion, the criteria for “relevant, irrelevant and additional records” and overall quality. In Figure 1 the number of healthy controls is not shown. The difference between pathological and phosphorylated α-synuclein remains unclear (Legend Table 1).
The results summarized in Table 1 and the Discussion page 6 do not refer to the working hypothesis outlined in lines 71 ff on page 2.
In the Discussion section page 6 a second literature search is briefly described immediately followed by results, but how these results were obtained remains open. Again, details of the literature search and the quality assessment of publications are missing. Which type of skin changes (in sun-exposed areas) are likely to be due to underlying α-synuclein depositions is unclear until the end of the manuscript. The discussion contains hypotheses and pathophysiological considerations (page 8) but does not refer to the findings on Table 1. On page 9, the discussion is followed by a second search, the methods of which are described in further detail. However, the applied criteria for the quality of retrieved publications (line 288) are missing.
In the Conclusions no specific results of the second literature search lines 239 ff are summarized and not discussed. Hormonal influences, medications and neuroinflammation are mentioned, but left without citation and reference to the search.
Author Response
We thank the reviewers for the careful reading of our manuscript and for the valuable comments, which helped us clarify several aspects that were missed or insufficiently explained in our original submission.
We have carefully considered all comments and revised the main text accordingly, using track changes to highlight the modifications.
In the document below, our point-to-point responses to the reviewers’s comments are in bold, and the passages shown in bold Italicscorrespond to text that has been directly incorporated into the revised manuscript.
We hope that the revisions and clarifications provided have satisfactorily addressed all concerns and have strenghtened the manuscript.
Reviewer 2
The topic is interesting and of differential diagnostic relevance in early α-synucleinopathies.
Thank you.
Abstract: Alzheimer is not topic of the manuscript and should therefore not be highlighted in the abstract. Skin changes in sun-exposed areas of the skin are addressed in the abstract, but findings in relation to PD and other α-synucleinopathies found in the literature search are not described but hypothesized. The abstract does not sufficiently reflect and summarize the contents of the manuscript.
We revised the abstract to ensure consistency with the content of the manuscript.
Manuscript: The methods of the literature research one (Table 1) are not described (pubmed? Scopus? Google?...Key words?). Not described are the “predefined criteria” of inclusion and exclusion, the criteria for “relevant, irrelevant and additional records” and overall quality.
The journal format places the Materials and Methods section after the Results and before the Conclusions, so the structure of the article may appear a little unusual and confusing. All this information can be found starting on page 5 of the manuscript. We have highlighted the Materials and Methods section in yellow to symplify its identification.
In Figure 1 the number of healthy controls is not shown.
Figure 1 only reports the number of articles, as it is a PRISMA flow-chart. We have clarified in the Results section that the number of healthy controls is 136. Moreover, because Figure 2 focuses on patients only, with specification of their clinical condition, we have added another figure (Figure 2a) that includes both all patients and healthy controls for clarity.
The difference between pathological and phosphorylated α-synuclein remains unclear (Legend Table 1).
We clarified in the legend that we were referring to pathological alpha-synuclein, as stated in the main text.
The results summarized in Table 1 and the Discussion page 6 do not refer to the working hypothesis outlined in lines 71 ff on page 2.
You are right. In the first paragraph of the Discussion (now highlighted in yellow) we explicitily state that no studies analyzed skin spots. The theory we propose is therefore based on a possible mechanism rather than on direct observations from the available studies. In this paragraph, we also clarify that we can only speculate: “The collected data identified the presence of α-synuclein in patients with neurological diseases, but without any information about the presence of clinically visible skin spots in the individuals included in the study. For this reason, we can only speculate about a possible relationship between skin spots and α-synuclein”.
In the Discussion section page 6 a second literature search is briefly described immediately followed by results, but how these results were obtained remains open. Again, details of the literature search and the quality assessment of publications are missing.
We understand that the journal format, which places the Materials and Methods section after the Discussion and before the Conclusions, may disrupt the reading flow. We have highlighted in yellow this information in this revision to show that all relevant information was correctly included.
Which type of skin changes (in sun-exposed areas) are likely to be due to underlying α-synuclein depositions is unclear until the end of the manuscript.
In the Discussion, we clarified the expected aspect of the lesions. “…the expectation is that these spots are clearer than aging one with more irregular borders.” This corresponds to what we state in the Conclusions regarding the skin spots (highlighted in yellow in this revision): “Herein, we propose that the evaluation of paler, more irregularly pigmented spots….”.
The discussion contains hypotheses and pathophysiological considerations (page 8) but does not refer to the findings on Table 1.
Yes, in the first paragraph of the Discussion we clarify this point, stating: “The collected data identified the presence of α-synuclein in patients with neurological diseases, but without any information about the presence of clinically visible skin spots in the individuals included in the study. For this reason, we can only speculate about a possible relationship between skin spots and α-synuclein”.
On page 9, the discussion is followed by a second search, the methods of which are described in further detail.
The methods are described after the Discussion and before the Conclusions, as required by the journal format. This is the only Materials and Methods section present in the paper.
However, the applied criteria for the quality of retrieved publications (line 288) are missing.
We understand that the journal format, which moves the Materials and Methods section after the Discussion and before the Conclusions, may disrupt the reading flow. We have highlighted in yellow this information in this revision; you can find the information about the quality of the included papers at page 6 of our manuscript.
In the Conclusions no specific results of the second literature search lines 239 ff are summarized and not discussed. Hormonal influences, medications and neuroinflammation are mentioned, but left without citation and reference to the search.
We revised the conclusions that now state “The studies currently available in the literature have identified the presence of α-synuclein in the skin of individuals affected by α-synucleinopathies. However, we did not find any studies specifically evaluating skin spots in patients with neurological diseases”.
Moreover we also included “Although our literature review did not identify studies addressing skin spots, the proposed mechanistic hypothesis on the role that α-synuclein might have in skin cells provides a rationale for conducting targeted investigations on skin spots.” This reiterates that we only speculate about skin spots because there are currently no studies on this finding.
Finally we have added the missing references in the Conclusions.
Round 2
Reviewer 1 Report
Comments and Suggestions for Authors
The table title should be at the top of the table, see editorial requirements.
Comments on the Quality of English LanguageEnglish is currently better.
Reviewer 2 Report
Comments and Suggestions for Authors
I suggest publication in the reviewed form.